# What Drives Climate Action in Canada's Provincial North? Exploring the Role of Connectedness to Nature, Climate Worry, and Talking with Friends and Family

Lindsay P. Galway [1,*], Thomas Beery [2,3], Chris Buse [4] and Maya K. Gislason [5]

1   Department of Health Sciences, Lakehead University, Thunder Bay, ON P7B 5E1, Canada
2   Faculty for Natural Sciences, Kristianstad University, SE-291 88 Kristianstad, Sweden; thomas.beery@hkr.se
3   Faculty for Teacher Training, Kristianstad University, SE-291 88 Kristianstad, Sweden
4   Centre for Environmental Assessment Research, University of British Columbia, Kelowna, BC V1V 1V7, Canada; cbuse@mail.ubc.ca
5   Faculty of Health Sciences, Simon Fraser University, Burnaby, BC V5A 1S6, Canada; maya_gislason@sfu.ca
*   Correspondence: lgalway@lakeheadu.ca

**Abstract:** Despite widespread calls to action from the scientific community and beyond, a concerning climate action gap exists. This paper aims to enhance our understanding of the role of connectedness to nature in promoting individual-level climate action in a unique setting where climate research and action are lacking: Canada's Provincial North. To begin to understand possible pathways, we also examined whether climate worry and talking about climate change with family and friends mediate the relationship between connectedness to nature and climate action. We used data collected via postal surveys in two Provincial North communities, Thunder Bay (Ontario), and Prince George (British Columbia) (*n* = 628). Results show that connectedness to nature has a direct positive association with individual-level climate action, controlling for gender and education. Results of parallel mediation analyses further show that connectedness to nature is indirectly associated with individual-level climate action, mediated by both climate worry and talking about climate change with family and friends. Finally, results suggest that climate worry and talking about climate change with family and friends serially mediate the relationship between connectedness to nature and with individual-level climate action. These findings are relevant for climate change engagement and action, especially across Canada's Provincial North, but also in similar settings characterized by marginalization, heightened vulnerability to climate change, urban islands within vast rural and remote landscapes, and economies and social identities tied to resource extraction. Drawing on these findings, we argue that cultivating stronger connections with nature in the places where people live, learn, work, and play is an important and currently underutilized leverage point for promoting individual-level climate action. This study therefore adds to the current and increasingly relevant calls for (re-)connecting with nature that have been made by others across a range of disciplinary and sectoral divides.

**Keywords:** climate action; connectedness to nature; climate worry; provincial north; mediation



## 1. Introduction

The scientific community has clearly articulated the need for climate action to limit the rise in global average temperatures to 1.5 °C above pre-industrial levels and cope with the current and "locked in" consequences of climate change [1,2]. Diverse, widespread, and immediate climate action (i.e., efforts to reduce greenhouse gas emissions, strengthen resilience, and build adaptive capacity to climate impacts [3]) is imperative in this pursuit. Despite a clear consensus that climate change is an emergency and a clarion call to action from the scientific community and beyond, a concerning climate action gap persists [4]. In this paper, we examined the role of connectedness to nature in promoting individual-level climate action in a unique setting where climate research and climate action are

lacking: Canada's Provincial North. The Provincial North is a region characterized by marginalization, heightened vulnerability to climate change, urban islands within vast rural and remote landscapes, and economies and social identities tied to resource extraction [5–7].

Our focus on connectedness to nature as a possible driver of individual-level climate action is informed, in part, by our interest in identifying opportunities for promoting climate action in ways that can also confer ancillary benefits to people and places; a co-benefits approach [8]. Based on existing evidence illustrating positive and robust associations between nature connection and environmentally responsible behaviors [9,10], alongside the compelling evidence that connectedness to nature promotes physical and mental health [11], quality of life [12], and environmental stewardship [13,14], promoting nature connectedness in the places where we live, learn, work, and play may be a unique leverage point to galvanize individual climate action in alignment with a co-benefits approach [14,15].

It is important to note that although this study focuses on individual-level climate action specifically, we recognize that individual action is a necessary but insufficient response to climate change and understand individual climate action as a socially embedded and place-based process [16,17]. Organizational, institutional, and systemic action are also essential to tackling climate change effectively, rapidly, and equitably. Nevertheless, individual-level action, particularly in Canada which consistently ranks among the top three nations globally for per capita emissions and is among the least likely of G2O countries to achieve 2030 emissions-reductions targets, is a necessary component of a comprehensive climate strategy [18,19].

Against this backdrop, we add to the existing literature aimed at identifying opportunities for promoting individual-level climate action from a co-benefits perspective by examining the following question: *Does connectedness to nature drive individual-level climate action in Canada's Provincial North?* Moreover, informed by recent calls to examine the processes that mediate the relationship between nature connectedness and individual action across diverse contexts [10], we also examined two potential mediators: climate worry and talking about climate change with family and friends. This research extends the existing literature on the relationship between connectedness to nature and individual-level climate action in three ways. First, we examined this relationship using a connectedness to nature in place perspective. Second, we tested indirect pathways examining climate worry and talking about climate change with family and friends as potential mediating factors. Third, we examined these relationships in an understudied region and context.

Below, we provide relevant background information on connectedness to nature in place, climate worry, and interpersonal climate communication. We then provide a summary of our hypotheses, describe the study communities specifically, and within the regional Provincial North context, and present the data collection and analysis methods utilized to achieve our objectives. Finally, we discuss findings in relation to existing literature and highlight future research directions to further advance our understanding of individual-level climate action in relation to nature connectedness.

### 1.1. A Connectedness to Nature in Place Perspective

The environmental connectedness perspective hypothesizes that spending time in nature will, given repeated experience, help individuals feel connected to nature, take notice of environmental change, be more inclined to care about nature, and, ultimately, increase the likelihood that individuals will engage in action to protect nature. According to Beery and Wolf–Watz [20] the phrase "environmental connectedness perspective" refers to the range of related theories that describe affective, cognitive, and physical human relationships with nature by using terms such as affinity, biophilia, commitment, ecological self, identity, inclusion, relatedness, sensitivity, and topophilia [21–33]. Within this broad grouping, the focus is placed on the experience of and direct encounter with nature and the possible affective, cognitive, and physical relationships between an individual and nature that strengthens from these experiences. However, this proposed progression from

experience-to-action, similar to the oversimplified environmental knowledge-to-action sequence [34], may not be as linear as originally hypothesized with many other variables coming into play along complex pathways, and likely varying across cultures and places. In considering how people experience nature and the experience-to-action progression, Beery and Wolf–Watz [20] argue that people experience nature and conceive of nature in place. In other words, people experience nature in the places where they live, learn, work, and play. We fine-tune this idea and propose situating the study of connectedness to nature in place, specifically, in this research, in two regional centers in the Canadian Provincial North.

Place is commonly understood to include three components: geographic location, material form, and cultural and subjective meaning (i.e., a sense of place [35]). One of the earliest descriptions of place attachment by Low (1992), describes it as bonding between people and places. Low's idea is very similar to affective dimensions of connectedness to nature theory [27]. People assign meaning to spatially and/or physically identifiable areas making them social, personal, and valued (i.e., assigning subjective meanings). Given contested ideas of nature [36], grounding connectedness to nature in place also moves beyond dualistic divisions of nature/culture or nature/people; place offers the potential for relational understanding where people and their environments are products of their various connections [37,38].

Support for this alignment between place theory and connectedness to nature can also be found in the research literature. Basu et al. [39] describe the positive and significant relationship between nature connectedness and place attachment, demonstrating a directly proportional relationship between the two constructs. Basu et al. [39] also show that higher levels of cognitive, emotional, and experiential connection with nature increases place dependence and develops place identity and nature bonding. Relatedly, research exploring the connection between "objects of care" and climate change [40] indicates that places can be the cherished objects of care that motivate action to address climate change when confronted by negative change or the potential for change and loss [41]. We are reminded by Galway [6] that just as people experience nature in places, climate change is a global phenomenon that is experienced and understood in local and regional places. Moreover, the need for, and value of, place-based climate change research and action is increasingly recognized [6,7,42].

In this current study, we argue for and apply a Connectedness to Nature in Place perspective (hereafter C2NP). We ground this approach using the literature on place attachment while also drawing inspiration from the concept of topophilia [43]. *Topophilia* was inspired by the evolutionary biological connectedness idea of *biophilia* [33]. Wilson [33] defined biophilia as an innate affinity of life or living systems. Sampson [43] intended topophilia to go beyond the living and included physical, nonliving elements of nature. While broadening biophilia, he also limited topophilia to the idea of affiliation with distinguishable places; place as specific rather than general or nebulous. Topophilia provides definitional clarity for the idea of human affiliation with the non-human world and more explicitly allows for a hybridized explanation of connectedness with place that includes experience, cultural learning, and innate-based origins [21,43].

The question of whether connectedness to nature shows a relationship with environmentally responsible behavior (i.e., pro-environmental behavior) or climate action specifically is, as previously noted, complex. Nonetheless, when the complexity of variables is considered, the relationship is positive [44–46]. Likewise, when place is used as part of a consideration of connectedness to nature [47,48] empirical research shows a relationship between connection to place and pro-environmental behaviors. Notably, a meta-analysis of studies examining the relationship between nature connection and pro-environmental behavior provides compelling evidence for positive associations across different operationalizations of nature connection, across different pro-environmental behavior measures (i.e., behavioral intentions, self-reports of behavior, and observed behavior), and across various sample and demographic characteristics [10]. However, this existing research also

emphasizes the need to examine this relationship across different contexts and cultures and to consider possible processes and mechanisms which may modify or mediate this underlying relationship [10].

*1.2. The Mediating Role of Climate Worry*

Empirical research examining the emotional impacts of, and responses to, climatic and environmental change has grown over the past two decades. Negative emotions including worry, anger, sadness, guilt, helplessness, and fear are increasingly associated with climate change [41]. This body of research has inspired and informed the development of a constellation of related concepts to describe, talk about, and study the affective dimensions of environmental degradation. In his book "Earth Emotions: The Words for a New World" Albrecht [49] outlines and describes a range of negative emotions related to climate change and environmental degradation as "psychoterratic emotions". Climate worry is a psychoterratic emotion; it is a specific form of worry associated with climate change that involves negative imagery, thought, and talk about climate change and the likely consequences of climate change [50,51].

Survey data consistently illustrate that people are increasingly worried about climate change and the present and future impacts of climate change; the prevalence of climate worry is on the rise [50,52]. Although climate worry can lead to psychological distress and health consequences, we argue that climate worry is often an appropriate, generative, activating, and adaptive response to climate change. It is not surprising that climate worry is increasing as the evidence documenting the consequences of climate change also grows, the likelihood of keeping global mean temperatures below 1.5 degrees of warming diminishes and people increasingly experience extreme weather events such as heatwaves, floods, and wildfires [1]. Climate worry and associated psychoterratic emotions such as ecological grief [53], climate anxiety [54], and solastalgia [49,55] have also become a focus of the mainstream media.

Worry involves imagery, thought, and/or talk about future events with uncertain outcomes and the possibility of negative consequences [51,56,57]. Worry is negatively affect-laden and relatively uncontrollable and is generally understood as both a cognitive and emotional state [56,58]. Worry has been associated with significant implications for general well-being and a range of specific mental and physical health endpoints [58]. Conceptually, worry is related to, though distinct from, concern, fear, and anxiety [41]. Compared to fear and concern, worry is "relatively personal and active" and consequently more likely to motivate action and problem-solving processes [50,59]. Worry is considered a primary cognitive characteristic of anxiety and is common across many anxiety disorders [60]. Worry is also recognized as being related to hope in that both emotions center around future-oriented thinking. Unlike worry, "hope is imbued with a positive feeling about the future; it is a kind of a conviction about the unproven" [61]. Some scholars and practitioners argue that climate worry, in the absence of hope, may ultimately lead to impacts on mental and emotional wellbeing and/or inaction, particularly among youth and young adults [62,63].

Despite the growing prevalence of and interest in climate worry, there has been limited empirical research examining climate worry generally, the specific antecedents of climate worry, or the influences of climate worry on support for climate policy and climate action [40,41,64]. Some recent evidence indicates that climate worry is an important predictor of public support for climate policy and may influence both collective and individual level climate action [40,50,59,65–68]. Using nationally representative survey data from the United States, Smith and Leiserowitz [59] showed that worry about climate change is a more important predictor of support for national climate and energy policies than sociodemographics, worldview, or other emotions. A recent and robust study by Bouman et al. [50], using survey data from 44,387 respondents across 23 European countries, generated strong evidence of the role of climate worry in motivating support for specific climate policies and individual-level climate mitigation.

Currently, numerous knowledge gaps remain in terms of understanding how climate worry influences climate action. Moreover, the causes and implications of climate worry are likely place-based and shaped by particular geographical, historical, and cultural contexts emphasizing the need for research across a range of settings [40,41,67]. Consequently, identifying what generates climate worry in specific places, may help to provide a better understanding of public support and engagement with the issue. As Bouman et al. [50] write "there is still much unknown about the origins and outcomes of worry about climate change" in particular, with respect to pathways and processes. Here, we argue that C2NP acts as an origin of climate worry, which in turn influences individual climate action.

*1.3. The Mediating Role of Talking about Climate Change with Family and Friends*

Although the public now generally recognizes climate change as an important issue, one that is largely human-caused, and one that is associated with a range of consequences for ecosystems, infrastructure, and human well-being, people rarely talk about climate change in their daily lives. In other words, despite rising levels of knowledge and awareness about climate change and increasing levels of climate worry, a "climate of silence" persists in many cultures and places [69]. Research shows that interpersonal communication about climate change is low across a range of settings [70–72]. Here, we define interpersonal communication as a "situated social process in which people who have established a communicative relationship exchange messages to generate shared meanings and accomplish social goals" [73].

Marshall [72] argues that a socially constructed silence surrounds climate change resulting in minimal interpersonal communication about climate change. Specifically, people rarely discuss the issue in their everyday lives or within social and familial circles. For example, according to a nationally representative survey in 2020, approximately two-thirds of Americans report talking about climate change with their family and friends rarely or never [70]. Research from the Yale Program on Climate Change Communication has shown that interpersonal communication around climate change can shift public opinion about the issue, and thereby could also influence individual-level climate action [74]. Discussions about climate change with family and friends may be particularly influential given that friends and relatives are often trusted sources of information [75]. Interestingly, using nationally representative survey data (from the United States), Hannibal et al. [76] found that discussions about climate change with family may influence climate change policy preferences, independent of perceived and assessed knowledge about climate change.

Although the research on climate change communication has grown substantially over the past three decades, few studies have specifically examined interpersonal communication [74,75]. We address this knowledge gap by examining the role of talking about climate change with family and friends as a mediator in the relationships between C2NP and individual climate action. Specifically, we argue that C2NP may lead people to talk more about climate change with family and friends which in turn may inspire or motivate people to engage in climate action.

*1.4. Overview of Hypotheses*

Together, the existing and emerging research summarized above, key knowledge gaps, and our interests in promoting place-based climate action in Canada's Provincial North have inspired our interest in exploring the relationship between C2NP and climate action and the role of climate worry and talking about climate change as possible mediating factors. To the best of our knowledge, there is no previous empirical research examining climate worry and talking about climate change as mediators between connectedness to nature and individual level-climate action. The present study adds to the literature addressing predictors of individual-level climate action in Canada and within the Provincial North context specifically.

Against this backdrop, this study considers the following four hypotheses (see Figure 1 for an overview):

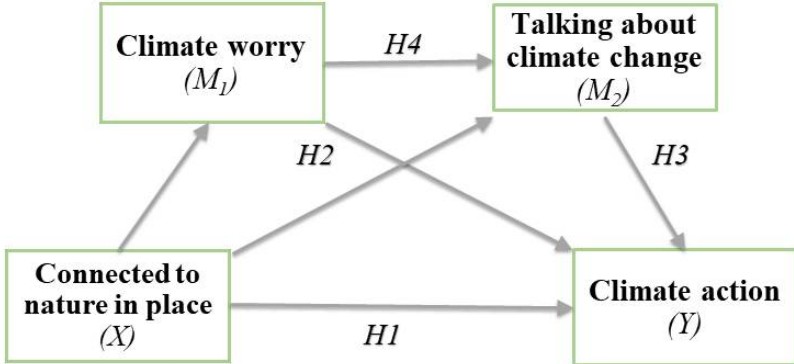

**Figure 1.** Summary of hypotheses (source: figure created by authors, adapted from [77]).

**Hypothesis 1 (H1).** *People with greater C2NP are more likely to engage in individual-level climate action.*

**Hypothesis 2 (H2).** *The relationship between C2NP and individual-level climate action is partially mediated by climate worry.*

**Hypothesis 3 (H3).** *The relationship between C2NP and individual-level climate action is partially mediated by talking about climate change with family and friends.*

**Hypothesis 4 (H4).** *The relationship between C2NP and individual level climate action is serially mediated by climate worry and talking about climate change with family and friends.*

*1.5. Study Context*

This study was one specific aspect of a larger project which examined how to communicate climate change impacts and solutions to promote engagement with climate action in the context of Canada's Provincial North. The larger project, entitled *"Climate Change Communication and Engagement in Canada's Provincial North"*, used a place-based approach and employed mixed-methods in the context of two regional case-study settings of the Provincial North: Northern BC and Northern Ontario. Canada's Provincial North "is a regional band that extends across Canada from British Columbia to Labrador, and encompasses numerous traditional Indigenous territories, vast landscapes, diverse ecosystems, and deep wilderness" punctuated by regional centers and industrial camps [7]. Development across the region has historically been driven by resource extraction led by large non-local corporations as well as transportation [78,79]. Often called Canada's "forgotten north", the Provincial North is characterized by remoteness, political marginalization, underemployment, diverse Indigenous populations, resource–dependence, and heightened vulnerability to climate change [5]. However, Indigenous and settler peoples residing in this region also exhibit resistance and resilience to ongoing disturbances and boom and bust economies alongside strong connections with the land and nature.

The qualitative dimension of the *"Climate Change Communication and Engagement in Canada's Provincial North"* project included in-depth interviews with climate champions and interactive workshops to develop climate change communication strategies aimed at promoting citizen engagement with climate action in the context of Canada's Provincial North (findings are reported in Gislason et al. [7]). The quantitative dimension of the project used postal surveys to document public beliefs, attitudes, and perceptions of climate change and climate action. The postal surveys were implemented in two communities within the regional case-study setting: Thunder Bay (Northern Ontario) and Prince George (Northern BC). This paper uses data collected via these postal surveys to test our hypotheses and advance our collective understanding of the relationship between connectedness to nature and individual-level climate action.

Thunder Bay and Prince George

Thunder Bay is a remote midsize city located on the shores of Lake Superior in Northern Ontario, Canada and the traditional territory of Fort William First Nation. It is a small urban island in a vast and sparsely populated landscape. Thunder Bay has a population of approximately 121,000, a population that has been in decline over the past decade due to an ageing population and outmigration of young adults, and is a regional service center for rural, remote, and First Nations communities across Northern Ontario [80–82]. "The history, landscape, and culture of Thunder Bay were largely shaped by resource booms including the fur trade in the 17th century, the Western Wheat Boom until the First World War, and the pulp and paper industry more recently" [6]. Currently, the community has a more diversified economy based on the industries of pulp and paper, manufacturing, telecommunications, research and development, mining, and nature-based tourism. Thunder Bay is a close-knit community characterized by pride in the Boreal landscape and its location on the shores of the world's largest freshwater lake. The culture of the community is shaped by this natural landscape; a large proportion of the population has close relationships with nature and engage in an array of land-based recreational and subsistence activities.

Communities in Northern Ontario, including Thunder Bay, are vulnerable to the impacts of climate change and have already begun to experience changes and impacts associated with climate change. In Thunder Bay, winter temperatures have increased 1.2 °C since 1891, along with a shorter winter season [83]. Lake Superior's water temperature increased 2.5 °C between 1979–2006, and ice coverage has declined in the lake by 42% since the 1970s [84]. Extreme weather including extreme rain and flooding events, ice storms, wind storms, and heat waves have already impacted the city and Northern Ontario region. Climate change projections show that, by 2050, the average annual temperature in Ontario will increase by about 3.6 °C, with more significant warming expected and an increase in the frequency and intensity of extreme events in Northern Ontario and Thunder Bay (Ontario 2016).

Prince George is located at the confluence of the Nechako and Fraser rivers near the geographic center of the province of British Columbia and is often considered the unofficial capital of Northern BC as the region's most populous community. Prince George had a population of 86,622 according to the 2016 Canadian Census [85], and like Thunder Bay, is the major service center in an otherwise sparsely populated region. While the Lheidli T'enneh have resided in this region since time immemorial, Prince George was introduced as a formal western settlement and fur trading post in the early 19th century. The trading post of Fort George was established as a result, and industries such as agriculture and forestry became commonplace to the region by the early 1900s. The Grand Trunk Pacific railway ultimately positioned Prince George as a major shipping and trade route between the coast and other parts of Canada. Prince George continues to build upon its longstanding forestry industry, although other industries such as healthcare, oil and gas, and even mining are headquartered or have regional offices in the city.

Because Northern Canada is warming at a rate faster than the rest of Canada, Prince George is also being notably impacted by climate change. Annual temperatures are projected to increase by 1.6–2.5 °C by the 2050s, and precipitation is expected to increase by 3–10% [86]. These warming patterns are projected to alter stream flows across the northern interior region where Prince George is located and alter weather patterns. Perhaps most significantly given the region's reliance on forestry is that climate change has warmed winter temperatures and enabled the survival and spread of the Mountain Pine Beetle and Spruce Beetle which have devastated 9 million hectares and 1.5 million hectares of surrounding forest, respectively, over the past 20 years [87], as well as threaten salmon populations.

## 2. Materials and Methods

### 2.1. Data Collection and Sample

This research was approved by the Research Ethics Boards at Lakehead University, the University of Northern British Columbia, and Simon Fraser University prior to data collection. As described above, the data used in the present study were collected via a postal survey as part of a larger community-based and mixed-method project. The postal survey was developed and administered in Thunder Bay and Prince George to gather quantitative data regarding beliefs, attitudes, and perceptions of climate change and climate action among citizens and to inform the development of strategies and programs to inform community-based climate communication and action.

Instrument development was informed by surveys previously developed and tested by the Yale Program on Climate Change Communication [88] and other validated scales, in collaboration with members of a Research Advisory Group. Specifically, the Research Advisory Group reviewed and provided comments on the instrument drafts to enhance relevance for the study setting. The survey instrument was also pilot tested with 19 community members prior to administration to ensure clarity of questions, estimate the total time to complete the survey, and identify any issues prior to administration. The final instrument consisted of 36 questions using a combination of Likert scale, ranking, fixed-choice answers, and open-ended questions in five main categories: (i) perspectives on climate change in general; (ii) climate change impacts; (iii) climate action; (iv) connectedness to nature; (v) demographic questions. The instrument is available from authors on request.

The postal survey was distributed by Canada Post mail to 4000 randomly selected households—2000 in Prince George and 2000 in Thunder Bay (using the Census Metropolitan Area as a sampling frame, populations of 86,622 and 121,621, respectively)—to gather data from a representative cross-section of adults. A simple random selection of households was selected from all addresses using the Canada Post address database.

The Dillman Tailored Design Method [89] was adapted to increase response rates and involved three waves of mailing. On 4th January 2019, survey packets containing the survey instrument, an information letter, and a pre-paid envelope to return the completed survey were sent out. The information letter explained the survey and encouraged participation by an adult member of the household, age 18 or older. If there was more than one adult in the household, instructions indicated that the person who has had the most recent birthday should complete the enclosed survey. The letter also included information about a random draw ($100 gift card) for those who completed the survey to enhance the response rate. On 18th January, a first reminder postcard was sent. On 25th January, the second and final reminder postcard was sent, along with information about how to complete the survey electronically. Both reminder postcards also included information about how to get another survey packet if it was never received or was lost.

A total of 192 surveys did not reach selected households and were returned to sender by Canada Post. A total of 693 surveys were completed (76 were completed electronically). After an initial review of the data for missingness, 38 responses were excluded as a result of being incomplete (more than 50% of responses left blank) or duplicate entries resulting in an adjusted response rate of 17.2%. Data from the postal surveys were entered into an Qualtrics electronic database by two trained research assistants to reduce the likelihood of data entry errors. Additionally, the final dataset was reviewed after data entry for any errors.

The final sample was composed of a total of 628 participants and reflected the total number of respondents for which there were no missing values in the regression analysis model. The majority of the sample was female ($n = 368$, 56%) and the median age was 60 years (SD = 17). Most of the sample either have children ($n = 468$, 72%) or would like children in the future ($n = 132$, 20%). Education levels ranging from primary school only (3%) to graduate-level degree (11%) with most respondents spread relatively evenly between. The education group with the most representation was those participants with a university certificate and the bachelor level or above (23%). The sample slightly overrepre-

sents older female adults, 55 years of age and older specifically, compared to 2016 census data for the study communities which may have implication for our findings.

*2.2. Measures*

2.2.1. Predictor Variables

*Connectedness to nature in place (C2NP).* The C2NP variable was measured using a scale developed from a review of connectedness to nature and place attachment literature [24,26–28,30,31,90–95]. The scale draws from both connectedness to nature and place attachment measures, yet with a deliberate nature-in-place identification, i.e., "the natural environment in *Thunder Bay (Prince George) and the surrounding region*." An effort was made to design a scale that was localized to be specifically useful to the Provincial North setting; this effort is in line with calls for broadening research methodologies in place-based and nature connectedness research [96]. The scale was derived from a composite of five items. Participants were asked to indicate their level of agreement with the following statements:

1. I am very attached to the natural environment in Thunder Bay/Prince George and the surrounding region;
2. I would feel less attached to Thunder Bay/Prince George and the surrounding region if the native plants and animals that live here disappear;
3. I learn a lot about myself when spending time in the natural environment in Thunder Bay/Prince George and the surrounding region;
4. When I spend time in the natural environment in Thunder Bay/Prince George and the surrounding region, I feel at peace with myself;
5. When I spend time in the natural environment in Thunder Bay/Prince George and the surrounding region, I feel a deep sense of oneness (i.e., connectedness) with the natural environment.

The items were measured using a five-point Likert-type scale ranging from 1 (strongly disagree) to 5 (strongly agree). The scale shows good face validity when compared to other connectedness to nature and place attachment scales [27,28,90,91,93–95]. Reliability testing was conducted as a part of a preliminary analysis; the five items comprising the proposed criterion variable of connectedness to nature/place were initially tested for internal consistency or reliability. The results, a Cronbach's alpha of 0.81, showed consistent response and indicated high intercorrelation. Results of the reliability test support the idea that the five items from the scale all measure the same latent construct.

2.2.2. Mediator Variables

*Climate worry.* Respondents were asked, "How worried are you about climate change?" on a 4-point scale (1 = not at all worried, 2 = not very worried, 3 = somewhat worried, 4 = very worried). This item was selected as it has been validated and consistently used in the Yale Program on Climate Communication surveys.

*Talking about climate change with family and friends.* Respondents indicated how often they talk about climate change with family and friends on a 4-point scale: never, rarely, occasionally, or very often. This item has also been validated and consistently used in the Yale Program on Climate Communication surveys.

2.2.3. Outcomes Variable: Individual-Level Climate Action

The climate action variable was measured using a scale developed from 14 survey questions asking respondents about a set of specific individual-level behavior and action items. The set of specific items included in the survey were largely informed by consultation and expertise of community partners involved in the research and intended to capture climate change-related action relevant to the study communities. In addition to their place-based grounding, the items show construct validity based on similarities with scales used in other environmental behavior investigations [97] and with those directed specifically at climate change response behavior [62,98].

Respondents indicated their participation (yes/no) with diverse forms of individual-level climate action items such as whether or not they have recycled, reduced beef consumption, weatherized their home, joined/donated to/volunteered with an organization working on issues related to climate change, made their views on climate change clear to politicians/decision-makers, and other specific action. A Cronbach's alpha test of the scale, $\alpha = 0.63$ indicates an acceptable level of reliability [99]. The items were calculated for a total action, or behavior, score with each individual item contributing 1 or 0 (corresponding to yes or no responses). The total action score was then re-coded into a scale with values 1–5.

*2.3. Analyses*

Our hypotheses were tested using PROCESS, a modelling program in SPSS that enables the exploration of parallel, moderated, and serial mediation models [100]. Specifically, PROCESS Model 4 assesses parallel mediation and PROCESS Model 6 assesses serial mediation [100,101]. In parallel mediation, two or more variables (M1, M2, etc.) are proposed to mediate the relationship between X and Y (see Figure 2) [102]. Serial mediation tests a pathway linking hypothesized mediators with a specified direction (X→ M1→ M2→ Y)) [100].

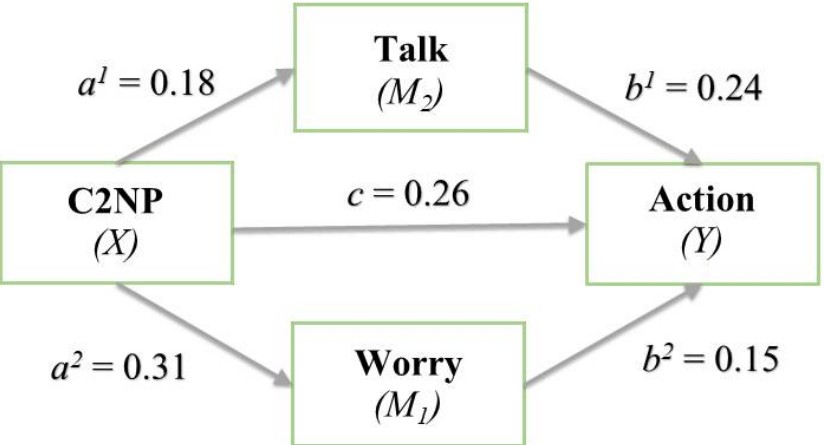

**Figure 2.** The mediating effects of climate worry and talking about climate change with family and friends in the relationship between connectedness to nature and climate action (source: figure created by authors, adapted from [77]).

In this study, a test of the direct effect of C2NP (X) on individual climate action (Y) was conducted; in addition, parallel mediation was used to test the mediating effect of climate worry (M1) and talking about climate change with friends and family (M2). We also tested for serial mediation, specifically that C2NP leads to more climate worry, which in turn leads to more talking about climate change with friends and family, ultimately leading to higher levels of climate action (i.e., C2NP → climate worry→ talking about climate change → climate action).

We included gender and education as covariates in the models to statistically account for possible confounding. A bootstrapping method was used to generate confidence intervals for the mediation effects; specifically, bias-corrected confidence intervals were generated based on 5000 bootstrap samples [100,103]. We used a bootstrapping method as it is considered the most powerful, most effective method to use with small samples, and the least vulnerable to type I errors [77,100,103]. All statistical analyses were conducted using SPSS statistical software.

**3. Results**

Descriptive statistics, correlations among study variables, and reliability coefficients are shown in Table 1. Moderate and significant positive correlations were detected in relation to all variables included in the models shown in Figures 2 and 3.

**Table 1.** Correlations, means, and standard deviations for all variables included in the study.

| Measure | Talk | Worry | Action | Mean | SD |
|---|---|---|---|---|---|
| C2NP | 0.183 *** | 0.273 *** | 0.317 *** | 4.04 | 0.70 |
| Talk | | 0.449 *** | 0.329 *** | 2.92 | 0.693 |
| Worry | | | 0.336 *** | 3.24 | 0.814 |
| Action | | | | 3.03 | 0.81 |

Note *** $p < 0.001$.

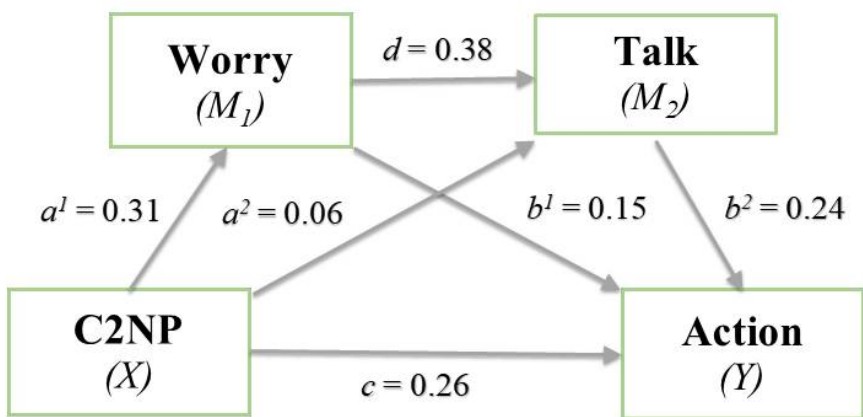

**Figure 3.** The serial mediating effects of climate worry and talking about climate change with family and friends in the relationship between connectedness to nature and climate action (source: figure created by authors, adapted from [77]).

Results from the parallel mediation analysis indicated that C2NP has a significant direct association with Action ($c = 0.26$, $p < 0.001$). Results also show C2NP is significantly and indirectly related to Action through the relationships with both Worry and Talk variables while controlling for gender and education. As can be seen in Figure 3, for those who report climate worry ($a_2 = 0.31$, $p < 0.001$), a higher level of worry was subsequently related to greater overall individual climate action ($b_2 = 0.15$, $p < 0.001$). A 95% bias-corrected confidence interval based on 5000 bootstrap samples indicated that the indirect effect through worry ($a_2b_2 = 0.05$, $p < 0.001$) holding all other mediators constant was entirely above zero (0.019 to 0.08).

The mediation model predicting Action from C2NP and Worry was significant ($F (1, 624) = 28.42$, $p < 0.001$), explaining 13% of the relationship (effect size calculations to interpret as the proportion of the total effect that is mediated were conducted given that the sample size is large and the total effect is larger than the indirect effect and of the same sign). In addition, as can be seen in Figure 2, for those who report talking about climate change with family and friends ($a_1 = 0.18$, $p < 0.001$), a higher level of talk was subsequently related to higher levels of action ($b_1 = 0.24$, $p < 0.001$). A 95% bias-corrected confidence interval based on 5000 bootstrap samples indicated that the indirect effect through talk ($a_1b_1 = 0.09$, $p < 0.001$) holding all other mediators constant, was entirely above zero (0.019 to 0.075). We found the mediation model predicting Action from C2NP and Talk was significant ($F (624) = 10.16$, $p < 0.001$), explaining 12% of the relationship.

Given the results of the parallel mediation, a serial mediation test was conducted. The overall regression model predicting Action from C2NP, through Worry and Talk was significant ($F (622) = 31.16$, $p < 0.001$), explaining 8% of the relationship (See Figure 3 and Table 2). Effect size calculations to interpret as the proportion of the total effect that is mediated were conducted given that the sample size is large and the total effect is larger than the indirect effect and of the same sign. Finally, we also ran the analyses disaggregated by community study (Thunder Bay and Prince George) and found that the results are consistent across the study communities in terms of direction and magnitude.

**Table 2.** Regression coefficients and standard errors for the serial mediation models in Figure 3.

| Predictor | Outcome | | | Talk | | | Action | | |
|---|---|---|---|---|---|---|---|---|---|
| | Worry | | | | | | | | |
| | Coeff. | SE | $p$ | Coeff. | SE | $p$ | Coeff. | SE | $p$ |
| C2NP | 0.31 | 0.04 | $p < 0.001$ | 0.06 | 0.04 | $p = 0.09$ | 0.26 | 0.04 | $p < 0.001$ |
| Worry | | | | 0.38 | 0.03 | $p < 0.001$ | 0.15 | 0.04 | $p < 0.001$ |
| Talk | | | | | | | 0.24 | 0.05 | $p < 0.001$ |
| | $R^2 = 0.12$ | | | $R^2 = 0.22$ | | | $R^2 = 0.20$ | | |
| | $F_{(624)} = 28.420, p < 0.001$ | | | $F_{(623)} = 43.98, p < 0.001$ | | | $F_{(622)} = 31.16, p < 0.001$ | | |

*Limitations*

There are several important limitations with respect to our study design and methods that must be recognized when interpreting findings. First, our study is correlational and therefore does not demonstrate causation. Second, given the cross-sectional design, the mediators have not been measured before the outcome variable such that we cannot confirm temporal precedence emphasizing again that we are unable to claim causal mechanisms. Future research using a longitudinal or sequential design (i.e., where C2NP is measured prior to hypothesized mediators, which in turn is measured prior to climate action) could address these limitations and provide evidence for causal relationships [104]. Third, although these findings are likely generalizable to people and places across Canada's Provincial North, in addition to communities in similar settings, they may not hold true in highly urbanized areas or in more rural and remote communities. Fourth, although our data are adequately representative of the study communities in terms of education and gender (as compared to 2016 census profiles), our data are not representative of the study communities in terms of race. Specifically, individuals self-identifying as Indigenous are underrepresented in our data. Additionally, our sample population somewhat overrepresents older adults (older than 55 years of age) which may have implications for our findings. Specifically, the findings reported herein may not be consistent in communities characterized by lower median ages. Fifth, youth were not included in our study sample such that we cannot confirm that our findings are illustrative of youth (younger than 18 years of age). Future research, working with youth and young adults, a unique population that stands to benefit the most from high levels of connectedness to nature, is therefore needed.

Although the items used to measure the mediator variables used in this study are robust in that they have been tested and used consistently by the Yale Program on Climate Communication program [88], there are also a set of limitations to note with respect to the measurement of mediator variables. Specifically, all of our measures are self-reported, which can threaten validity which can lead to inflated associations. The climate worry variables specifically were measured using a single-item rather than a multi-item scale. Research increasingly indicates that climate worry may be a multidimensional construct [51]. Recently (post-data collection in this study), Stewart [51] developed and tested a ten-item Climate Change Worry Scale which may benefit future research. With respect to our measurement of talk about climate change with friends and family, our variable does not indicate what people are talking about specifically, who initiated conversations, nor the relative influence of talking with friends versus family.

Finally, regarding our measurement of individual-level climate action, we opted to use community partner perspectives and experiences to ensure that our measurement of climate action was grounded in place. Although our measurement of climate action has not been previously validated, we find acceptable levels of validity and reliability [99]. Also, noteworthy, specific climate action items were measured as binary (yes/no) and therefore do not indicate frequency.

## 4. Discussion and Conclusions

This study provides support for C2NP as a driver of individual-level climate action, in the Provincial North of Canada specifically, while also expanding current understanding of possible pathways via climate worry and interpersonal communication. Using survey data collected in two communities in Canada's Provincial North we find that C2NP has a direct positive association with individual-level climate action; that is, those with greater self-reported C2NP are more likely to engage in individual-level climate action. C2NP is also indirectly associated with individual-level climate action, mediated by both climate worry and talking about climate change with family and friends. Additionally, using serial mediation analysis, our results show that those with higher levels of C2NP have higher levels of climate worry, which leads to talking about climate change with family and friends, which in turn leads to higher levels of climate action.

We want to emphasize that, based on the modest effect sizes and the complex interplay of factors that shape individual behavior, there are certainly other factors at play not captured in our models. However, we argue that cultivating stronger connections with nature in the places where people live, learn, work and play is an important and currently underutilized leverage point for promoting climate action. Therefore, our findings add to the current and increasingly relevant calls for (re-)connecting with nature that have been made by others across a range of disciplinary and sectoral divides [14,46,105–107].

*Implications and Future Research Priorities*

A disconcerting and compelling challenge of the Anthropocene is the existence of urgent converging crises such that synergistic solutions that offer co-benefits to people and places are imperative [19,108]. Following Abson et al. [96] and Ives et al. [14], we argue that fostering human–nature relationships should be recognized as a "deep" leverage point with the potential to offer a range of co-benefits beyond promoting climate action. Moreover, if we recognize that human disconnect from nature is a root cause and driver of the climate emergency and related environmental crises [109], it follows that (re-)connecting with nature should be at the heart of how we respond to, and address, climate change.

The global COVID-19 pandemic has highlighted the multiple values of nature in the places where we live, learn, work, and play [110–112]. Now is the time to identify, develop, and support human–nature relationships grounded in place-based approaches. Place-based approaches acknowledge that human–nature relations are shaped by the multiple dimensions of place, while also reflecting an understanding that climate action should be informed by, and grounded in, local and regional knowledges, experiences, landscapes, and priorities [7,42,55,113]. Programming and policy aimed at (re-)connecting people with nature should be recognized as a timely opportunity to inspire climate action, promote health and healing, motivate an ethic of care and reciprocity, and an entry point for considering the ongoing effects of settler-colonialism [114].

This study also provides evidence that higher levels of C2NP can lead people to feel more worried about climate change which, in turn, can inspire people to take action against climate change. Consequently, our findings add to the emerging literature illustrating that climate worry is an activating emotion; an emotion that moves us to take action in response to climate change [40,50,59,65,66,68]. Specifically, we add to this literature by illustrating that climate worry is an activating emotion when associated with C2NP. We suspect that having stronger connections with nature provides opportunities to understand, take notice of, witness, and/or experience the impacts of climate change which in turn inspires climate worry and action [115].

Drawing on our findings and the work of Wang et al. [40] we argue that when people have a strong sense of connectedness to nature in the places where they live, learn, work and play, nature and place can become "objects of care". Perceiving "objects of care" as threatened by climate change inspires strong emotional responses that can promote and sustain action [40,41]. This place-based emotion-to-action process can also be understood as reciprocity, a giving back based on a sense of belonging [116].

Like Van der Linden [67], we position climate worry as a central affective dimension with respect to climate change, particularly when it comes to understanding how emotion can motivate individuals to engage with climate action. However, as Albrecht [49] reminds us, our collective understanding of the affective dimensions of climate change is very much a work in progress as the "novel effects of grand-scale environmental damage to places, hearts, and psyches become evident" [49]. We agree with this assessment and point to several priorities for future research with respect to the emotional dimensions of climate change and climate action.

Additional research is needed to more fully understand when, where, for whom, and under which conditions climate worry is activating and constructive versus paralyzing and harmful. Although climate worry is an appropriate response when one understands, witnesses, and/or experiences climate change via C2NP, and, as we find, an activating emotion when associated with C2NP, climate worry may also become paralyzing and/or pathological leading to psychological distress and impacts on health and well-being for some people and populations [52,117]. Children and youth, which were not included in our study sample, are particularly vulnerable to experiencing psychological distress and health impacts in relation to climate worry underscoring the need for research focused on children and youth specifically [61,62]. Place-based, community-engaged, and participatory research with youth examining how climate worry is experienced, expressed, and managed is, therefore, a priority.

We also propose that future research should aim to unpack and more fully examine the nexus of climate worry, hope, and action. Emerging alongside the growing body of research examining climate worry is a growing body of work positioning hope as an activating emotion with respect to climate action [118,119]. Hope can "serve as a buffer and prevent worry from leading to low wellbeing or to undermining desire and ability to act" [120]. To date, there is a dearth of research examining the worry–hope–action nexus such that a range of unanswered questions remain [50]. Does C2NP also inspire a sense of hope about the future, in spite of the climate crisis? Do worry and hope together inspire action more so than climate worry alone? Is it that worry activates action while hope sustains it? What is the relationship between worry and hope in urban versus rural contexts?

Having a better understanding of the interplay between climate worry and hope, as they relate to C2NP and climate action, is an opportunity to prevent mental and emotional health consequences of climate worry and learn how to sustain climate action over the long run. In this space, there is a particular need for "grounded hope", which occurs when people realistically appraise the reality they find themselves in, grounded in the places where they live, learn, work, and play and decide to take action to make their lives, the lives of others better, and the prospects of future generations. At some point, most of us will face the task of recovering, rebuilding, and rebounding from adversity [121]. Grounded hope can help us to bounce forward, together. The interplay between worry, hope, and action also connects to what Ojala [61,118] calls meaning-focused coping; when people recognize the seriousness and urgency of climate change but "are able to activate positive emotions that can help them to bear the worry associated with the awareness of this threat" [61]. Future research aimed at developing and testing tools, programs, and processes that engage with this intersection of climate worry, hope, and action and are aimed at supporting meaning-focused coping is also an important direction for future research.

Finally, our findings have shown that C2NP influences interpersonal communication about climate change, among family and friends specifically, which in turn influences climate action. Our findings, therefore, emphasize the "power of conversations" [122]. As described above in relation to climate worry, we posit that feeling connected to nature in the places where we live, learn, work, and play, provides opportunities to understand, take notice of, witness, and/or experience the impacts of climate change which, in this case, inspires climate change conversations with family and friends. Overall, interpersonal communication is understudied and underutilized as a possible pathway supporting

climate action [75,76]. To confirm our findings and advance a collective understanding of the links between connectedness to nature, talking about climate change, and action, additional research in other contexts is needed. For example, our mediation analysis may demonstrate that people are having conversations about their climate worry, driving action. Although we are not able to confirm this, our findings point to the possibility that talking about climate worry may in fact be a specific mechanism to ensure that feeling worried about climate change is activating rather than harmful or paralyzing.

Macy and Johnstone [122] urge us to collectively recognize that "[t]he more we draw issues into the open, the more included we become to tackle them". Inspired by this perspective, we argue that there is a need to move climate conversations beyond the interpersonal realm and into broader public conversations about the future of our communities and the lands on which they are located. Diverse sectors, groups, and organizations are increasingly designing and creating tools and processes to motivate and facilitate conversations about climate change in community and in place. Notably, efforts at creating safe spaces for talking about climate change, including the emotional dimensions of climate change like climate worry, and efforts aimed at (re)connecting with nature can both be understood as "work that reconnects" [122]. Collectively engaging in the work that reconnects is perhaps our grand challenge and opportunity as we emerged from the COVID-19 pandemic and focus our gaze and efforts towards co-creating a healthy, just, and sustainable future.

**Author Contributions:** L.P.G., C.B., and M.K.G. designed the "Climate Change Communication and Engagement in Canada's Provincial North" project and survey instrument; L.P.G.; led the postal survey implementation and data collection; T.B. led data analysis and interpretation; L.P.G. and T.B. led the manuscript writing. All authors contributed to revisions and the final version of the manuscript. Authors are listed in order of contribution. All authors have read and agreed to the published version of the manuscript.

**Funding:** This research was funded by The Social Sciences and Humanities Research Council of Canada, grant number 1466071.

**Institutional Review Board Statement:** This study was approved by the Research Ethics Boards at Lakehead University, the University of Northern British Columbia, and Simon Fraser University prior to data collection (code: 1466071; Date of approval: 2017/12/04).

**Informed Consent Statement:** Informed consent was obtained from all participants involved in the study.

**Data Availability Statement:** We have not made these data publicly available as this was not articulated at the time of data collection nor was this approved in our REB process.

**Acknowledgments:** Thank you to all members of the Research Advisory Groups for their contributions and wisdom.

**Conflicts of Interest:** The authors declare no conflict of interest.

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
