# Peer review of "What Drives Climate Action in Canada’s Provincial North? Exploring the Role of Connectedness to Nature, Climate Worry, and Talking with Friends and Family"

_climate, doi:10.3390/cli9100146_

Round 1

Reviewer 1 Report

I have read the article which is quite interesting. My comments are:

  1. The authors should attach a copy of the questionnaire to the manuscript.
  2. The authors should also state their position in regards to data availability.
  3. The regression model should and estimation should be properly specified.

Reviewer 2 Report

- The title is long, but concrete and necessary. I do have some doubts about whether it is best to start with a question. This question, perhaps, hinders the visibility and readability of the text in title and keyword searches.
- The abstract is clear, concise, and contains all the expected parts. I sincerely congratulate the authors. It is commendable that at the end, they have not failed to highlight why their text represents an advance for the scientific community, as should be the case at the end of any good abstract. 
- I think there are some paragraphs that are too long. It would be good to achieve a more harmonious distribution of paragraphs, with a more visual reading, and that they do not contain more than 5-7 lines each paragraph. This clarifies ideas and makes the text more readable. 
- The first time an acronym appears, the first letter should be capitalised: page 3, lines 139 and 140; page 9, line 444. 
- Review and remove any double blanks, e.g. page 5, line 241. 
- The source of the images is missing, even if they are self-made (this should be indicated). 
- Review the repetition of Thunder Bay, which appears too many times in a row, on page 7. Perhaps pronouns could be used to clean up the text a little and make the reading more concise and with less repetition. 
- The methodology is clear and well explained, although it would be good to add some more references to the work (not in the theoretical framework, but in the methodological section). 
- The results are well ordered and explained, although it is necessary to put the font on all the images and perhaps add colour or make them more aesthetic. They can be replicated by other researchers and it would be good if they were more striking and suggestive. 

Reviewer 3 Report

The paper is valuable in understanding how people reaction to the climate change issues. The majority data is female and median age is 60. One suggestion is to clean the data similar to the region demographic description.  
